MYB transcription factors in alfalfa (Medicago sativa): genome-wide identification and expression analysis under abiotic stresses

Zhou Qiang
Jia Chenglin
Ma Wenxue
Cui Yue
Jin Xiaoyu
Luo Dong
http://orcid.org/0000-0002-9899-0246 Min Xueyang
Liu Zhipeng lzp@lzu.edu.cn
State Key Laboratory of Grassland Agro-ecosystems, Key Laboratory of Grassland Livestock Industry Innovation, Ministry of Agriculture and Rural Affairs, Engineering Research Center of Grassland Industry, Ministry of Education, College of Pastoral Agriculture Science and Technology, Lanzhou University , Lanzhou , China
Hoecker Ute
Electronic publication date: 2019 Sep 17
Publication date: 2019
Volume: 7
Electronic Location ID: e7714
Received 2019 Apr 18; Accepted 2019 Aug 21
Copyright: © 2019 Zhou et al.
Copyright year: 2019
Copyright holder: Zhou et al.
License: This is an open access article distributed under the terms of the Creative Commons Attribution License, which permits unrestricted use, distribution, reproduction and adaptation in any medium and for any purpose provided that it is properly attributed. For attribution, the original author(s), title, publication source (PeerJ) and either DOI or URL of the article must be cited.
License URL: https://creativecommons.org/licenses/by/4.0/

Keywords: Medicago sativa, MYB, Transcription factor, Expression profiles, Abiotic stress

Funding: National Natural Science Foundation of China, Grant/Award Numbers 31722055 and 31672476 This research was supported by the National Natural Science Foundation of China, Grant/Award Numbers 31722055 and 31672476. The funders had no role in the study design, data collection and analysis, decision to publish, or preparation of the manuscript.

==============================
Background

Alfalfa is the most widely cultivated forage legume and one of the most economically valuable crops in the world. Its survival and production are often hampered by environmental changes. However, there are few studies on stress-resistance genes in alfalfa because of its incomplete genomic information and rare expression profile data. The MYB proteins are characterized by a highly conserved DNA-binding domain, which is large, functionally diverse, and represented in all eukaryotes. The role of MYB proteins in plant development is essential; they function in diverse biological processes, including stress and defense responses, and seed and floral development. Studies on the MYB gene family have been reported in several species, but they have not been comprehensively analyzed in alfalfa.

Methods

To identify more comprehensive MYB transcription factor family genes, the sequences of 168 Arabidopsis thaliana, 430 Glycine max, 185 Medicago truncatula, and 130 Oryza sativa MYB proteins were downloaded from the Plant Transcription Factor Database. These sequences were used as queries in a BLAST search against the M. sativa proteome sequences provided by the Noble Research Institute.

Results

In the present study, a total of 265 MsMYB proteins were obtained, including 50 R1-MYB, 186 R2R3-MYB, 26 R1R2R3-MYB, and three atypical-MYB proteins. These predicted MsMYB proteins were divided into 12 subgroups by phylogenetic analysis, and gene ontology (GO) analysis indicated that most of the MsMYB genes are involved in various biological processes. The expression profiles and quantitative real-time PCR analysis indicated that some MsMYB genes might play a crucial role in the response to abiotic stresses. Additionally, a total of 170 and 914 predicted protein–protein and protein-DNA interactions were obtained, respectively. The interactions between MsMYB043 and MSAD320162, MsMYB253 and MSAD320162, and MsMYB253 and MSAD308489 were confirmed by a yeast two-hybrid system. This work provides information on the MYB family in alfalfa that was previously lacking and might promote the cultivation of stress-resistant alfalfa.

Introduction

Alfalfa (Medicago sativa) is the most widely cultivated forage legume, and one of the most economically valuable crops in the world (Liu et al., 2013). In China, alfalfa plantation areas are distributed in 14 provinces throughout the northern region of the country. Freezing temperatures, deficient water, and high salinity are major factors affecting alfalfa growth and limiting its productivity and survival throughout this region. Therefore, cultivation of alfalfa germplasm with high stress resistance will play an important role in the development of animal husbandry in Northern China. However, there are few studies on stress-resistance genes in alfalfa because of its incomplete genomic information and rare expression profile data. In order to maintain proper homeostasis for normal growth, plants have evolved multiple ways to combat harsh environments by mobilizing a wide spectrum of stress responsive genes (Muvunyi et al., 2018). For example, the MYB (V-myb avian myeloblastosis viral oncogene homolog) family of transcription factors (TFs) are known to play defensive roles in plants during abiotic stresses.

The MYB family of TFs is named for its conserved MYB domain (DNA-binding domain), and is large, functionally diverse, and represented in all eukaryotes. The MYB domain is generally composed of one to four imperfect repeats (R) in plants. Therefore, the MYB family is classified into four subfamilies based on the number and position of repeats, namely, R1-MYB, R2R3-MYB, R1R2R3-MYB, and atypical-MYB (Yanhui et al., 2006; Jin & Martin, 1999). Compared with animals and yeasts, the structures and functions of MYB TFs are more conserved in plants (Li, Ng & Fan, 2015). The role of MYB proteins in plant development is essential; they function in diverse biological processes, including stress and defense responses, regulation of primary and secondary metabolism, seed and floral development, and cell fate and identity determination (Li, Ng & Fan, 2015). For example, an MYB family transcription factor circadian1 (AtCIR1) increased tolerance to freezing stress before and after cold acclimation by increased expression of CBF genes, which indicated that AtCIR1 positively regulates cold responsive genes and cold tolerance (Guan et al., 2013). A 1R-MYB protein AtMYBC1 was shown to be a repressor of freezing tolerance in a CBF-independent pathway (Zhai et al., 2010). Ectopic expression of the wheat MYB TF TaMYB31-B gene in Arabidopsis thaliana affected plant growth and enhanced drought tolerance by up-regulation of wax biosynthesis genes and drought-responsive genes (Zhao et al., 2018). Overexpression of ThMYB13 in Tamarix hispida led to a low level of reactive oxygen species (ROS) and a stable K+/Na+ ratio, which indicated that ThMYB13 might play a role in T. hispida during salt stress (Zhang et al., 2018b). Transgenic A. thaliana plants overexpressing ZmMYB3R displayed increased sensitivity to abscisic acid (ABA), and enhanced tolerance to drought and salt stress via an ABA-dependent pathway (Wu et al., 2019). The MYB TFs can simultaneously respond to multiple abiotic stresses. A previous research project reported that GmMYB118 could improve tolerance to drought and salt stress in soybean plants through promoting expression of stress-associated genes and regulating osmotic and oxidizing substances to maintain cell homeostasis (Du et al., 2018). Furthermore, overexpression of another gene, MYB49, in tomato plants was associated with significant tolerance to drought and salt stresses (Cui et al., 2018). Therefore, the MYB gene family acts as an important TF to improve plant resistance to abiotic stress.

To date, studies on the MYB gene family have been reported in many plants. A total of 104, 166, 155, 197, 524, and 475 MYB genes of all the types were identified and analyzed in Lotus japonicas, M. truncatula, Oryza sativa, A. thaliana, Gossypium hirsutum, and Brassica rapa ssp. Pekinensis, respectively (Katiyar et al., 2012; Saha et al., 2016; Salih et al., 2016; Wang & Li, 2017). Moreover, 244 and 155 R2R3-MYB genes were reported in Glycine max and M. truncatula (Du et al., 2012; Zheng et al., 2017), respectively. These reports are favorable resources for the study of the MYB gene family. However, although the MYB gene family acts as an important TF to improve plant resistance to abiotic stress, only one research paper on the MYB gene family has been published relating to alfalfa (Dong et al., 2018). In this study, two alfalfa cultivars, Dryland and Sundory (SD), which differed with respect to their ability to tolerate salinity stress, were sequenced to investigate participation of MYB TFs in the salinity stress of alfalfa. A total of 17 MYB TFs were isolated and analyzed, and it was found that MsMYB4 significantly increased the salinity tolerance of alfalfa in an ABA-dependent manner. These results provide a valuable resource for future studies on the MYB gene family in alfalfa. The first alfalfa genome data at the diploid level (CADL) were released by the Noble Research Institute in 2017, and this provides alfalfa researchers with important research resources. In order to identify more comprehensive MYB TF family genes in alfalfa, a total of 168 A. thaliana, 430 Glycine max, 185 M. truncatula, and 130 O. sativa MYB proteins were used as queries, using the cultivated alfalfa at the diploid-level genome blast server. Moreover, we analyzed the responses of some MsMYB genes to ABA treatment and cold, drought, and salt stresses. The results from this study will provide new information on the evolution of the MYB family proteins and protein structures, and valuable information for further studies of MYB genes in alfalfa.

Materials and Methods

Identification of the MYB gene family in alfalfa

To identify more comprehensive MYB TF family genes, the sequences of 168 A. thaliana, 430 Glycine max, 185 M. truncatula, and 130 O. sativa MYB proteins were downloaded from the Plant Transcription Factor Database (http://planttfdb.cbi.pku.edu.cn/) (Jin et al., 2017). These sequences were used as queries in a BLAST search against the M. sativa proteome sequences provided by the Noble Research Institute (https://www.alfalfatoolbox.org/) (Altschul et al., 1990), with an E-value cut-off of 0.00001. The obtained MYB sequences were confirmed based on the presence of intact MYB domains using the Pfam Program (http://pfam.xfam.org/), and the expectation cut-off (E value) 1.0 was set as the threshold for significance (El-Gebali et al., 2019). Moreover, the remaining sequences were analyzed by the cluster database at high identity with tolerance (CD-HIT) web server (http://www.bioinformatics.org/cd-hit/) (Li & Godzik, 2006), using default parameters to remove redundant data. Subsequently, non-redundant sequences were renamed as predicted MsMYB genes. The grand average of hydropathicity (GRAVY) index values and theoretical isoelectric point (pI) of these predicted MsMYB proteins were determined by the ProtParam Tool (https://web.expasy.org/protparam/) (Gasteiger et al., 2003). Additionally, subcellular localization of MsMYB genes was predicted using the Target P 1.1 server (Emanuelsson et al., 2000), and validated in WoLF-PSORT (https://www.genscript.com/wolf-psort.html) (Horton et al., 2007).

In silico functional analysis of MsMYB genes

The MsMYB protein function was predicted by (gene ontology) GO annotation, using the web-accessible Blast2GO v4.1 annotation system (https://www.blast2go.com/) (Gotz et al., 2008). Briefly, the MsMYB protein sequence was used to search for similar sequences against the NCBI non-redundant (Nr) database using the Blast tool in the Blast2GO software, with an expectation value of 10−3. Next, mapping and annotation were performed on Blast2GO using default parameters. Moreover, GO functional classification was performed by WEGO 2.0 (Ye et al., 2018). In addition, GO enrichment analysis for MsMYB genes was conducted on agriGO v2.0 (http://systemsbiology.cau.edu.cn/agriGOv2/) with default parameters (Tian et al., 2017), and the A. thaliana gene model (TAIR9) was selected as the reference.

Conserved motif and phylogenetic analysis of the MsMYB genes

In order to analyze the sequence features of MYB repeats in R2R3-MYB proteins, the amino acid sequences of R2 and R3 repeats of all R2R3-MYB proteins in M. sativa were extracted, and multiple sequence alignments of these identified R2R3-MYB proteins were performed using MUSCLE with default parameters. The sequence logos for R2 and R3 repeats were generated using the WebLogo (http://weblogo.berkeley.edu/logo.cgi) with default settings (Crooks et al., 2004). In addition, the motifs conserved among MsMYB members were identified using the multiple expectation maximization for motif elicitation (MEME) v4.11.1 online program (http://meme-suite.org/index.html) (Bailey et al., 2009). The maximum number of motifs was set to 45, according to a previous report (Zheng et al., 2017), and other parameters used the default settings. Moreover, to clarify the evolutionary relationship of these predicted MsMYB TF family genes, multiple sequence alignments of MYB protein sequences were performed using Clustal W (http://www.clustal.org/clustal2/) with default parameters (Larkin et al., 2007). The subsequent phylogenetic analysis relied on the neighbor-joining (NJ) method, as implemented in the MEGA v6.0 software (https://www.megasoftware.net/) (Tamura et al., 2013), and a bootstrap analysis was applied based on 1,000 replicates. Additionally, nonsynonymous (Ka) and synonymous (Ks) substitution rates were calculated to explore the mechanism of gene divergence after duplication, and Ka and Ks were computed using DnaSP 5 software (Librado & Rozas, 2009).

In silico expression analysis of the MsMYB genes during plant development

Genome-wide transcriptome data from M. sativa in different tissues during development were downloaded from the CADL-Gene Expression Atlas (https://www.alfalfatoolbox.org/atlasCADL/) provided by the Noble Research Institute (O’Rourke et al., 2015). The transcriptome data were derived from six tissues, including leaf, flower, pre-elonged stem, elonged stem, root, and nodule. Subsequently, these expression data were analyzed and clustered using the hierarchical cluster program MEV 4.9.0 (http://www.mybiosoftware.com/mev-4-6-2-multiple-experiment-viewer.html) to draw a heatmap of MsMYB genes in different tissues during development.

In silico transcriptome analysis of the response of the MsMYB genes to abiotic stress

In previous studies, a total of four transcriptome sequencing projects were performed to obtain additional genetic information for alfalfa relating to the response to abiotic stress, including cold (SRR7091780–SRR7091794, Zhou et al., 2018), and ABA, drought, and salt treatments (SRR7160313–SRR7160357, Luo et al., 2019a, 2019b). In this study, the expression level of some MsMYB genes (with the absolute value of fold change ≥ 2) was obtained after the local nucleotide blast (BLASTN) against these four transcriptome data was performed (Camacho et al., 2009). The clustering analysis and the heat map generated were performed by the hierarchical cluster program MEV 4.9.0.

In silico analysis of cis-regulatory element

Sequences of 2,000 bp from promoters of these changed MsMYB genes during abiotic stresses were analyzed for potential cis-regulatory elements, by querying them through the PlantCARE database (http://bioinformatics.psb.ugent.be/webtools/plantcare/html/). A total of six cis-regulatory elements were recorded, including abscisic acid responsive (ABRE), CGTCA-motif (methyl jasmonate responsive), G-box (light inducible), low-temperature responsive (LTR), MBS binding site (drought responsive), and TC-rich repeats (defense and stress responsive).

Plant materials and stress treatments

The plant materials used were alfalfa variety Zhongmu No. 1, which was cultivated and provided by the Qingchuan Yang Laboratory of the Beijing Institute of Animal Sciences, Chinese Academy of Agricultural Sciences. The experimental samples in this study were obtained by a hydroponic experiment. Before germination, vernalization of seeds was conducted at 4 °C to maintain the consistency of germination. After 3 days of seed germination, the seedlings were transferred into a nutrient solution (1/2 MS, pH = 5.8) and grown under a 16 h light/8 h dark cycle at 22 °C. Different stress treatments of seedlings were performed when the third leaf of the alfalfa appeared (approximately 10 days after germination). For cold treatment, the seedlings were placed in an artificial climate incubator and frozen under a 16 h light/8 h dark cycle at 4 °C. Four cold-treatment time points were used (2, 6, 24, and 48 h), along with one control. In the drought or salt treatments, the seedlings were transferred into a nutrient solution containing 400 mM Mannitol or 250 mM NaCl, and grown under a 16 h light/8 h dark cycle at 22 °C. There were eight treatments in these two experiments, which included seven treatment time points (1, 3, 6, 12, 24 h, and stress removal 1 and 12 h) and one control. For ABA treatment, the seedlings were transferred into a nutrient solution containing 10 μM ABA, and a total of three ABA treatments were performed, including 1, 3 and 12 h treatment time points. In order to be consistent with the experimental materials of transcriptome sequencing in our laboratory and obtain tissues closely related to abiotic stress, the whole seedling was harvested for cold treatment, and the root tip was harvested for the ABA, drought, and salt treatments. A total of six seedlings were collected and pooled into a frozen tube pipe, one for each treatment at the corresponding time point, and these samples were flash-frozen in liquid nitrogen and stored at −80 °C.

Quantitative real-time PCR analysis

Ribonucleic acid was extracted from the whole seedlings (cold treatment) or root tips (ABA, drought, and salt treatments) of control and treated seedlings using the Trizol method (Sangon Biotech, Shanghai, China), according to the manufacturer’s instructions. The concentration of each sample was determined using a NanoDrop ND1000 spectrophotometer (Thermo Scientific, Waltham, MA, USA). A one μg aliquot of DNase-digested total RNA was used to synthesize the single-strand cDNA using a FastKing RT Kit (Tiangen Biotech, Beijing, China), following the manufacturer’s protocol. The subsequent qRT-PCR was performed using a TB Green™ Premix Ex Taq™ Kit (TaKaRa, Dalian, China) on a CFX96 Real-Time PCR Detection System (Bio-Rad, Los Angeles, CA, USA). Each 20 μL reaction contained 10 μL TB Green Premix, 0.2 μM of each primer, and two μL cDNA. The reactions were initially denatured (95 °C/30 s), then subjected to 40 cycles of 95 °C/5 s, 60 °C/30 s. According to the transcriptome sequence of Zhongmu No. 1, gene-specific primers were designed using Primer Premier 6 software (Premier Biosoft International, Palo Alto, CA, USA), and are shown in Table S1. Each of the three biological replicates were supported by three technical replicates, and the relative expression levels were normalized to the expression of the Medicago actin gene (AA660796) (Liu et al., 2016, 2017a) and calculated using the 2−ΔΔCt method.

Protein–protein and protein-DNA interactions predictions

The predictions of protein–protein and protein-DNA interactions for MsMYB genes were carried out by the online server Arabidopsis Interactions Viewer (http://bar.utoronto.ca/interactions/cgi-bin/arabidopsis_interactions_viewer.cgi), which queries a database of 70,944 predicted and 39,505 confirmed A. thaliana-interacting proteins. The expression patterns of genes that interact with MsMYB genes during abiotic stresses were analyzed using the hierarchical cluster program MEV 4.9.0, and the correlation coefficient between the expression levels of the MsMYB gene and its interacting genes was calculated. In addition, the MYB-core motifs (C/TNGTTA/G) were searched in the upstream sequence (2,000 bp) of the homologous gene of interacting genes in M. truncatula, and these interacting genes were obtained by the prediction of protein-DNA interactions. To verify the reliability of these predicted protein–protein interactions, a total of five predicted protein–protein interactions were selected (Table S2). The recombinant pLexA and pB42AD plasmids were co-transformed into yeast strain EGY48 and plated on medium lacking Ura, His, and Trp (-U-H-T) at 30 °C. The colonies were transferred to induction medium lacking Ura, His, Trp, or Leu (-U-H-T+X-gal and -U-H-T-L+X-gal) for interaction screening.

Results

Identification and classification of the alfalfa MYB gene family

To identify MYB genes in the M. sativa genome, a BLASTP search was performed using 168 A. thaliana, 430 Glycine max, 185 M. truncatula, and 130 O. sativa MYB protein sequences as queries. Then, a Pfam search was used to ensure that they contained the MYB DNA-binding domain, for verification of the identity of these MYB sequences. Additionally, these putative MYB genes were inspected using CD-HIT to ensure they possessed complete open reading frames and maps to unique genomic loci. As a result, a total of 265 non-redundant MYB sequences were obtained and named MsMYB001 to MsMYB265, based on the order of their serial number in the database of the M. sativa proteome sequences. These MYB TFs were classified into four distinct groups, namely, “R1-MYB,” “R2R3-MYB,” “R1R2R3-MYB,” and “atypical-MYB” genes, based on the presence of one, two, three, or four MYB repeats, respectively. Our analysis revealed that the R2R3-MYB subfamily consisted of the highest number of MYB genes, and the number of these four types of MYB genes was 50 (18.87%), 186 (70.19%), 26 (9.81%), and 3 (1.13%), respectively (Table 1). The classification information of each gene is shown in Table S3. In a previous study, 17 MsMYBs were predicted and analyzed (Dong et al., 2018). In order to verify the accuracy of predicted MsMYB genes in our study, a BLAST search for 17 reported MsMYBs in the 265 predicted MsMYBs was performed. As a result, 15 of the 17 reported MsMYBs were found in these 265 MsMYBs, and the details are shown in Table S4. However, the other two reported MsMYBs were not found, which may be due to the inconsistencies in the sequencing of samples between these two sequencing projects provided by Dong et al. (2018) and the Noble Research Institute.

Table 1 The MYB-domain based characterization and comparison of MYB transcription factor family genes in terms of GRAVY and molecular weight.

MYB groups	No. of genes	Length (aa)	Molecular weight (D)	PI	GRAVY	
Min.	Max.	Avg.	Min.	Max.	Avg.	Min.	Max.	Avg.	Min.	Max.	Avg.	
R1	50	79	1,424	281	9,130.13	155,631.15	31,348.92	4.39	11.37	7.87	−1.001	−0.024	−0.634	
R2R3	186	103	1,050	337	11,811.84	118,277.10	38,204.10	4.41	9.93	7.03	−1.101	−0.177	−0.725	
R1R2R3	26	166	1,879	604	19,320.12	208,302.91	68,200.79	4.92	10.06	7.09	−0.999	−0.290	−0.795	
Atypical	3	478	1,433	1,052	54,100.16	156,566.27	117,915.47	5.17	8.74	6.81	−0.691	−0.459	−0.591	
All	265	79	1,879	361	9,130.13	208,302.91	40,756.14	4.39	11.37	7.19	−1.101	−0.024	−0.713	

In addition, the physiochemical properties of these MsMYB proteins were also analyzed, including protein length, molecular weight, pI, GRAVY, and subcellular localization (Table S3). The 265 predicted MsMYB proteins ranged from 79 (MsMYB259) to 1,879 AA (MsMYB173) in length, with an average length of 360.76 AA (Table 1). The molecular weight of these MsMYB proteins ranged from 9,130.13 D in MsMYB047 to 208,302.91 D in MsMYB173. Moreover, the mean of pI and GRAVY was 7.19 and −0.71, respectively. We also predicted the subcellular localization of these MsMYB proteins using several localization predictor software programs. Our analysis revealed that 232 (87.55%) MsMYB proteins were found to be nuclear-localized (Table S3). The remaining 33 MsMYB proteins were predicted to be localized in the chloroplast (15), cytoplasm (11), plasma membrane (2), vascular bundle (2), endoplasmic reticulum (1), golgi apparatus (1), and mitochondria (1).

In silico functional classification of MYB transcription factors

For the functional annotation of all MsMYB genes, GO functional analysis was performed by the Blast2GO program. In this study, a total of 23 GO categories were assigned to the 265 MsMYB genes (Fig. S1; Table S5). “Cell” gene (62, 23.4%) was the dominant category in the cellular component category, followed by “cell part” and “organelle” (61 for both, 23.0%). In the molecular function category, a total of 235 (88.7%) genes were assigned to “binding,” including “organic cyclic compound binding” (234, 88.3%), “heterocyclic compound binding” (234, 88.3%), and “ion binding” (4, 1.5%). Regarding the biological process category, “cellular process” (50, 18.9%) and “metabolic process” (48, 18.1%) were the most dominant groups. Additionally, we performed a GO enrichment analysis of the functional significance, using the agriGO website to identify the significantly enriched GO terms among these 265 MsMYB genes, with a p score cut-off of 0.05. As a result, a total of eight GO terms were considered to be significantly enriched among these genes (Table 2), and four and three GO terms belonged to “molecular function” (F) and “biological process” (P), respectively. However, only one GO term was significantly enriched in “cellular component” (C), namely “nucleus” (GO:0005634).

Table 2 Gene ontology (GO) enrichment analysis of 265 MsMYB genes.

GO term	Ontology	Description	Gene number	P-value	FDR	
GO:0034641	P	Cellular nitrogen compound metabolic process	47	4.9E-39	3.1E-37	
GO:0006807	P	Nitrogen compound metabolic process	47	0.0000058	0.00018	
GO:0051276	P	Chromosome organization	9	0.000013	0.00027	
GO:0003677	F	DNA binding	234	8.9E-254	1.9E-252	
GO:0003676	F	Nucleic acid binding	234	4.8E-203	5.2E-202	
GO:0005488	F	Binding	235	2.7E-121	2E-120	
GO:0016746	F	Transferase activity, transferring acyl groups	7	0.0049	0.026	
GO:0005634	C	Nucleus	60	1.4E-18	3.5E-17	

Phylogenetic analysis of MsMYBs

In this study, we performed phylogenetic analysis of M. sativa and M. truncatula MYB proteins using the NJ method to evaluate the evolutionary significance of the MYB genes. The circular phylogenetic tree was constructed based on the alignment of the amino acid sequences of 310 MYB genes, including 265 from M. sativa and 45 from A. thaliana, L. japonicas, and M. truncatula (15 genes for each species) (Table S6). Based on the topology and robustness of the NJ phylogenetic tree results in M. truncatula (Wang & Li, 2017), we resolved these 265 MsMYB genes into 12 subgroups (A–L), which ranged in size from three (H) to 52 (L) (Fig. 1).

Figure 1 Phylogenetic analysis and subgroup classifications of 265 MsMYB proteins and 45 MYB proteins from Arabidopsis thaliana, Lotus japonicus and Medicago truncatula.

The red, filled triangle denotes MYB genes in alfalfa; the blue, inverted triangle denotes MYB genes in A. thaliana; the green square denotes MYB genes in M. truncatula; and the purple diamond denotes MYB genes in L. japonicus. The genes in subgroups A, D, E, and G mainly respond to plant hormones; the genes in subgroups C and K mainly respond to abiotic stresses and plant hormones; the genes in subgroups A and B participate in the synthesis of secondary metabolites; the genes in subgroups F, I, and J participate in the development process; the genes in subgroup H respond to abiotic stresses and plant hormones, and participate in the development process; and the genes in subgroup L are only involved in transcriptional regulation.

In silico analysis of the conserved MYB domains in MsMYB genes

In order to investigate the sequence features of R2R3-MYB proteins in M. sativa, multiple sequence alignments were performed with Clustal W, and a highly conserved 104 AA region among all R2R3-MYB proteins was identified. Subsequently, the WebLogo program was used to study the variation within the conserved motifs of R2R3-MYB genes in alfalfa. As a result, the R2 and R3 MYB repeats of the MsR2R3-MYBs contain five highly conserved Trp (W) residues that play a key role in sequence-specific binding of DNA (Fig. 2). Of these five Trp residues, the third conserved tryptophan residue (W46) in the R2 repeat was not completely conserved in all of the MsR2R3-MYBs. Moreover, we also found that some amino acids showed high conservativeness, such as Glu (E), Asp (D), Leu (L), Arg (R), Lys (K), Ser (S), Cys (C), Gly (G), and Asn (N). Furthermore, the protein motif organization among all 265 MsMYBs was investigated using MEME. The MEME results showed that the width of 45 identified motifs range from 14 to 50 AA, and the maximum E-value of these motifs was 1.0e-041 (Table S7). In order to validate the clade definition based on phylogenetic analysis, the protein motif distribution for each MsMYB was analyzed. As a result, one or more motifs outside of the MYB domain were shared by most members within the same clade, which provides further support for the results of the phylogenetic analysis. The distribution of the conserved motifs within each clade is shown in Fig. S2. In order to explore the evolutionary patterns, divergence and selection pressure of the MsMYBs, a total of 44 homologous pairs were identified using phylogeny-based and bidirectional best-hit methods. The Ka/Ks ratio is widely applied to measure genetic evolution and selection pressure. Scatter plot statistics showed that all gene pairs had evolved mainly under the influence of purifying selection except for three homologous pairs (MsMYB082/084, MsMYB235/253, and MsMYB256/257) (Fig. 3).

Figure 2 The R2 and R3 MYB repeats are highly conserved across all R2R3-MYB proteins in alfalfa.

The sequence logos of the R2 (A) and R3 (B) MYB repeats are based on full-length alignments of all alfalfa R2R3-MYB domains. The bit score indicates the information content for each position in the sequence. Triangles indicate the conserved tryptophan residues (Trp) in the MYB domain, and asterisks denote the conserved residues that are identical among all MsR2R3-MYBs.

Figure 3 Scatter plot statistics of Ka and Ks values among alfalfa.

The black, dotted line in slope one is used to show Ka/Ks = 1. The red dots indicate the different MsMYB genes.

Expression profile analysis of the MYB genes in M. sativa tissues

Microarray datasets from the alfalfa B47 genotype were downloaded from the CADL-Gene Expression Atlas and used to assess the transcript abundance profiles of MsMYB-encoding genes in six major tissues: leaf, flower, pre-elonged stem, elonged stem, root, and nodule. We only examined the transcript abundance of 211 MsMYB genes because the remaining 54 genes were not represented in the dataset. In order to visualize the datasets, an expression heat map was obtained by the program MEV 4.9.0. According to the expression patterns of these genes over six tissues of the alfalfa B47 genotype, 211 MsMYB genes were divided into eight subgroups, namely A to H (Fig. 4). Subgroup A contains eight genes, and they showed the highest transcript accumulation level in nodule tissue. Subgroup B includes 40 MsMYB genes, which showed the lowest transcript accumulation in the root and nodule. In contrast, some of these genes showed the highest transcript accumulation level in the leaf, flower, or pre-elonged stem. Moreover, a total of 37 MsMYB genes were steadily expressed in all of the six tissues tested, and these MsMYB genes belong to subgroup C. Approximately 33 of these 211 MsMYB genes (subgroup D) showed the highest transcript accumulation level in flower tissue, 10 (subgroup E) showed the highest transcript accumulation in leaf tissue, 35 (subgroup F) showed the highest transcript accumulation in root or nodule tissue, and 30 (subgroup G) showed the highest transcript accumulation in the stem, including pre-elonged and elonged stems. The remaining 18 MsMYB genes (subgroup H) showed the lowest transcript accumulation in the leaf and flower.

Figure 4 Heat map representation of the expression profiles of the MsMYBs among different tissues.

Higher and lower levels of transcript accumulation are indicated by red and green, respectively, and the median level is indicated by black. Microarray data were obtained from the CADL-Gene Expression Atlas database, and the heat map was generated using MEV 4.9.0. Subgroups A, B, D, E, F, G, and H showed the highest transcript accumulation level in nodule, root and nodule, flower, leaf, root or nodule, stem, and leaf and flower, respectively. The remaining genes of subgroup C were steadily expressed in all of the six tissues tested.

Expression analysis of MsMYB genes in response to abiotic stresses

In order to investigate the expression levels of MsMYB genes under abiotic stresses, local nucleotide blast (BLASTN) against four transcriptome datasets reported by our laboratory was conducted. As a result, we determined that the expression of 51 and 52 MsMYB genes changed significantly during cold and three other stresses, respectively, and these 51 genes were present in all four treatments. Because tissue samples used for cold sequencing differed from the other three stresses, the expression patterns of these genes during cold treatment were analyzed separately from the other three treatments. As shown in Fig. 5, a total of 51 MsMYB genes were divided into four subgroups, namely A–D. Compared to the control samples, 11 MsMYB genes (subgroup B) were inhibited, and the remaining genes were induced, during cold stress. Interestingly, a total of 21 genes belonging to subgroup A and C were induced earlier by cold stress than those in subgroup D (19 genes). Moreover, the expression patterns of 52 MsMYB genes during ABA, drought, and salt treatments were also analyzed. Our analysis revealed that these MsMYB genes were also divided into four subgroups: A, B, C, and D. As shown in Fig. 6, subgroups A, B, and D were highly expressed in all three treatments. Additionally, subgroup C was highly expressed in response to drought and salt stresses, but most genes were not expressed in response to ABA treatment.

Figure 5 Expression of 51 MsMYB genes in response to cold treatment.

Heat map showing the changes in expression level of these MsMYB genes at different time points after treatment with 4 °C (0, 2, 6, 24, and 48 h) in the whole seedling, and “CK” indicates 0 h. Microarray data were obtained from the reported study in alfalfa (Zhou et al., 2018). The MsMYB genes of subgroup B were inhibited during cold stress, and members of the subgroups A and C were induced earlier by cold stress than those in subgroup D.

Figure 6 Expression of 52 MsMYB genes in response to ABA, drought, and salt treatments.

Heat map showing the changes in expression level of these MsMYB genes at different time points after treatment with 10 μM abscisic acid (0, 1, 3, and 12 h), 400 mM mannitol (0, 1, 3, 6, 12, 24, removal 1 and 12 h) and 250 mM NaCl (0, 1, 3, 6, 12, 24, removal 1 and 12 h) in the root, and “CK” indicates 0 h. Microarray data were obtained from the reported studies in alfalfa (Luo et al., 2019a, 2019b). Subgroups A, B, and D were highly expressed in all three treatments. Subgroup C was highly expressed in response to drought and salt stresses, but most genes were not expressed in response to ABA treatment.

Cis-regulatory element in MsMYB gene promoters

Cis-regulatory elements control expression patterns of stress-responsive genes, and these elements are located upstream of gene-coding sequences and provide binding sites for TFs. Thus, we investigated the distribution of six cis-regulatory elements in 52 stress-responsive MsMYB gene promoters. As a result, a total of 107 ABRE cis-elements were identified, which was more than the number of the other five cis-elements (Fig. 7). Meanwhile, we found at least one cis-element in each MsMYB gene promoter, and only the MsMYB040 promoter contained all six cis-elements.

Figure 7 Cis-regulatory elements in the promoter regions of MsMYBs.

A colored block with a number represents the number of cis-elements in the analyzed promoter region of the indicated MsMYBs. Sequences of 2,000 bp from promoters of 52 MsMYB genes were downloaded from the alfalfa databases provided by the Noble Research Institute (https://www.alfalfatoolbox.org/).

Gene expression analysis qRT-PCR validation

To further confirm the RNA-seq responses of these 52 MsMYB genes to abiotic stresses, qRT-PCR was performed for 14 MsMYBs that were significantly induced genes under various stress treatments (Figs. 5 and 6; Table S1), including ABA, cold, drought, and salt. The expression patterns of most of the MsMYB genes in the qRT-PCR analysis were consistent with the RNA-Seq analysis, but the magnitude of the fold changes varied between RNA-seq and qRT-PCR experiments (Fig. 8). Moreover, these results showed that all 14 genes were induced to different degrees by abiotic stress.

Figure 8 Expression analysis of four genes during abiotic stresses, according to qRT-PCR and RNA-seq.

White bars represent the relative expression levels determined by qRT-PCR (left y-axis). Black bars indicate the transcript abundance change based on the Fragments Per Kilobase per Million Fragments mapped (FPKM) values of the RNA-Seq analysis (right y-axis). Error bars indicate standard errors of the means (n = 3). Whole seedling was used for cold treatment (E–H); root tips were used for ABA (A–D), drought (I–L), and salt treatments (M–P). “*” indicates significance at the 0.05 level.

Protein–protein and protein-DNA interactions predictions

As a TF, the MYB gene can regulate plant development and responses to various environmental changes by controlling the expression of downstream genes, and is also regulated by the upstream genes. Therefore, the prediction of protein–protein and protein-DNA interactions was performed in this study to identify some candidate genes that putatively interact with the MsMYB genes, and a total of 178 and 957 predicted interactions were recognized, respectively. Subsequently, these predicted interaction genes were screened with the absolute value of fold change ≥ 2 as the threshold. As a result, a total of 170 protein–protein and 914 protein-DNA interactions were predicted (Tables S8 and S9, respectively), and the correlations between the MsMYB gene and its putative interactor’s genes are shown in Tables S10 and S11, respectively. Of the 914 predicted protein-DNA interactions, only one homologous gene promoter of interacting genes in M. truncatula does not contain any MYB-core motifs (C/TNGTTA/G) (Table S9). Additionally, there were 27 and 105 more reliably predicted protein–protein (fold change ≥ 2, correlation coefficient ≥ 0.8) and protein-DNA (fold change ≥ 10, correlation coefficient ≥ 0.8) interactions, and the expression patterns of these predicted interaction genes were affected to varying degrees by abiotic stresses (Figs. 9 and 10; Figs. S3 and S4). We found that most predicted protein–protein and protein-DNA interactions showed significant positive correlations during ABA or cold treatments, but negative correlations under drought or salt stresses (Fig. 11; Fig. S5). Furthermore, the yeast co-transformants of all five predicted protein–protein interactions grew on SD/-U-H-T medium (Fig. S6), and three yeast co-transformants turn blue on SD/-U-H-T+X-gal and SD/-U-H-T-L+X-gal induction mediums (Fig. 12), indicating that MsMYB043 and MsMYB253 strongly interacted with MSAD320162, and MsMYB253 also interacted with MSAD308489. However, the remaining two predicted protein–protein interactions didn’t show a significant relationship.

Figure 9 Hierarchical clustering of interaction gene expression profiles during cold treatment.

These interaction genes were predicted by protein–protein interaction, and “CK” indicates 0 h. The expression levels of these interaction genes with the absolute value of fold change ≥ 2 and the correlation coefficient between MsMYB genes and its interaction genes was more than 0.8. Microarray data were obtained from the reported study in alfalfa (Zhou et al., 2018).

Figure 10 Hierarchical clustering of interaction gene expression profiles during ABA, drought, and salt treatments.

These interaction genes were predicted by protein–protein interaction, and “CK” indicates 0 h. The expression levels of these interaction genes with the absolute value of fold change ≥ 2 and the correlation coefficient between MsMYB genes and its interaction genes was more than 0.8. Microarray data were obtained from the reported studies in alfalfa (Luo et al., 2019a, 2019b).

Figure 11 Correlation analysis between the expression patterns of MsMYB genes and their interaction genes during abiotic stresses.

These interaction genes were predicted by protein–protein interaction. The expression levels of these interaction genes with the absolute value of fold change ≥ 2 and the correlation coefficient between MsMYB genes and its interaction genes was more than 0.8.

Figure 12 Validation of the predicted interacting proteins by the yeast two-hybrid system.

Empty pLexA plus pB42AD, empty pLexA plus pB42AD-MsMYB043, empty pLexA plus pB42AD-MsMYB253, pLexA-MSAD320162 plus empty pB42AD, and pLexA-MSAD308489 plus empty pB42AD were used as negative controls; pLexA-JAZ2 plus pB42AD-bHLH13 was used as a positive control. (A) (C) and (E) indicate the induction medium lacking Ura, His, Trp (-U-H-T+X-gal); (B) (D) and (F) indicate the induction medium lacking Ura, His, Trp, and Leu (-U-H-T-L+X-gal).

Discussion

The MYB genes comprise a large family of TFs that are ubiquitous to all plant species. The genome-wide analysis of MYB gene families has been widely performed in many species whose genomes have been sequenced (Deng et al., 2016; Li et al., 2016a). In the present study, a search for MYB genes in the alfalfa genome resulted in the identification of 265 members. The number of MsMYBs is greater than that in L. japonicas (104 MYBs), M. truncatula (166 MYBs), O. sativa (155 MYBs), and A. thaliana (197 MYBs), but less than that in Gossypium hirsutum (524 MYBs) and Brassica rapa ssp. Pekinensis (475 MYBs) (Katiyar et al., 2012; Saha et al., 2016; Salih et al., 2016; Wang & Li, 2017). Of these 265 MsMYBs, the R2R3-MYB family is the most abundant MYB TF, which is consistent with previous studies of many plants (Salih et al., 2016; Wang & Li, 2017; Du et al., 2012). These results indicate that the predicted MsMYBs identified in our research can be used as valuable resources for functional analysis of MsMYBs.

In total, 310 MYB proteins, which include 265 MsMYB and 45 MYB proteins from A. thaliana, L. japonicas, and M. truncatula, were grouped into 12 different subfamilies through phylogeny analysis. The phylogeny analysis results of most of the MYB proteins from M. truncatula are consistent with the previous report (Wang & Li, 2017), which showed that the results of phylogeny analysis in the present study have a high degree of credibility. M. truncatula is a diploid leguminous forage and is a close relative of M. sativa; thus, their corresponding homologous MYB genes have similar expression patterns. Further, MtMYBS1 (Medtr1g073170.1) was inducible by ABA, NaCl, and PEG 6000, and it enhances salinity tolerance when constitutively expressed in A. thaliana (Dong et al., 2017b). Interestingly, the homologous gene MsMYB145 of MtMYBS1 was also induced under these abiotic stresses (Fig. 6). However, a previous study showed that M. sativa and M. truncatula belong to different subgroups, namely “sect. Medicago clade” and “truncatula clade,” respectively (Yoder et al., 2013). Therefore, there is a difference between MsMYB and MtMYB genes. In the present study, a comparative analysis between 265 MsMYB and 185 MtMYB genes from the Plant Transcription Factor Database was performed (Table S12). Compared with MtMYB genes, MsMYB genes are greater in number, and the expansion of MsMYB genes appear in subgroups K and L. Moreover, GO annotation and enrichment analysis of these 265 predicted MsMYBs were performed, which indicated that a total of 235 (88.7%) genes were assigned to “binding,” and it was also considered significantly enriched among all MsMYBs (Fig. S1; Table 2). Similar results have been reported in Brachypodium distachyon (Chen et al., 2017b). Besides “binding,” many GO terms were enriched in “cellular component” and “biological process,” such as “nucleus” and “cellular nitrogen and nitrogen compound metabolic process” (Table 2). These results indicate that most of the MsMYB genes are involved in various biological processes.

The MYB proteins are characterized by a highly conserved DNA-binding domain (the MYB domain), and this domain generally consists of up to four imperfect amino acid sequence repeats (R) of about 52 amino acids, each forming three α-helices (Dubos et al., 2010). Moreover, the second and third helices of each repeat build a helix–turn–helix (HTH) structure with three regularly spaced tryptophan (or hydrophobic) residues, forming a hydrophobic core in the 3D HTH structure (Ogata et al., 1996). In previous studies, a total of six conserved Trp residues were found to be evenly distributed in the R2 and R3 domains of MYB proteins (Du et al., 2012; Katiyar et al., 2012; Wang & Li, 2017; Zheng et al., 2017). However, five Trp residues were found in the R2R3 domains at particular positions in our study, and only two Trp residues exist in the R3 domain of MsMYB proteins (Fig. 2), which is consistent with the results of previous reports on Zea mays, Pyrus bretschneideri, and Ginkgo biloba (Chen et al., 2017a; Li et al., 2016b; Liu et al., 2017b). Similar results have been reported in a previous study on alfalfa (Dong et al., 2018).

The expression of a large number of MYB genes is required when plants transition from the vegetative to the reproductive stage (Chen et al., 2017b). Previous analysis demonstrated that many MYB genes were reported to be involved in plant development, such as root hair formation and development (Tominaga-wada & Wada, 2014; Zheng et al., 2016), stem development (Chai et al., 2014; Zhu et al., 2013), the change of leaf color and leaf rolling (Chen et al., 2017b; Guan et al., 2019; Huang et al., 2013; Kanemaki et al., 2018; Zhang et al., 2016), and the development of the anther, pollen, petals, gynoeca, and pollen tube (Kasahara et al., 2005; Preston et al., 2004; Reeves et al., 2012). According to the CADL-Gene Expression Atlas, we found that a total of 17.5% MsMYB genes (37/211) were steadily expressed in all of the six tissues tested, whereas some other genes displayed a tissue-specific expression pattern (Fig. 4). For example, 33, 10, 35, and 30 MsMYB genes were found to be expressed mainly in the flower, leaf, root, or nodule and stem tissues, respectively. In M. sativa, there has been no report that has proven that MYB genes regulate development directly, but the expression pattern indicates that some MYB genes might regulate the development of alfalfa. For example, a previous study has shown that AtMYBH (AT5G47390.1) participates in the regulation of leaf senescence in A. thaliana (Huang et al., 2015b), and its homologous gene MsMYB195 shows the highest transcript accumulation in leaf tissue (subgroup B, Fig. 4).

The relevant role of the MYB genes during plant stress tolerance has been reported in earlier studies using various transgenic plants. For example, overexpression of the wild soybean R2R3-MYB TF GsMYB15 enhances resistance to salt stress in transgenic A. thaliana (Shen et al., 2018). On the contrary, the R-R-type MYB TF AtDIV2 has a negative role in salt stress, and is required for ABA signaling in A. thaliana (Fang et al., 2018). Similarly, transgenic A. thaliana overexpressing FtMYB13 had a lower sensitivity to ABA, and improved drought and salt tolerance compared to the wild type (Huang et al., 2018). In M. sativa, a previous study demonstrated that MsMYB4 significantly increased the salinity tolerance of A. thaliana in an ABA-dependent manner (Dong et al., 2018). The expression profiles of many MYB genes indicate their probable functions in response to abiotic stresses. For example, the expression of OsMYB511 and OsMYB2 is markedly induced by cold in rice (Huang et al., 2015a; Yang, Dai & Zhang, 2012), and the expression of MYB21 and MYB24 is rapidly induced by jasmonate in A. thaliana (Stracke, Werber & Weisshaar, 2001). In the present study, the expression pattern of 52 MsMYB genes, determined in previous studies during multiple abiotic stresses, was analyzed. As a result, these genes were found to be induced to varying degrees by abiotic stresses (Figs. 6 and 7). Moreover, the expression of 14 MsMYB genes was detected by qRT-PCR, and the results showed that all of these 14 MsMYB genes were significantly induced by ABA, cold, drought, or salt (Fig. 8). However, the magnitude of the fold changes varied between RNA-seq and qRT-PCR experiments, and this phenomenon also appeared in several previous studies (Cao et al., 2018; Dong et al., 2017a; Wang et al., 2015; Zhang et al., 2018a). This may be due to sample collection for previous RNAseq and qPCR experiments at different times, or differences in RNA extraction or cDNA synthesis. Additionally, alfalfa is different from A. thaliana and other model plants, and it is an obligate outcrossing, autotetraploid (2n = 4x = 32) plant. Thus, there is genetic diversity between individual plants of alfalfa (Liu et al., 2013; Zhou et al., 2014), which may be another important reason for inconsistencies between RNAseq and RT-qPCR data. A previous study reported that ABRE is the major cis-element for ABA-responsive gene expression (Yamaguchi-Shinozaki & Shinozaki, 2006). A total of 107 ABREs from these 52 MsMYB genes were found in this study, and many cis-elements associated with abiotic stress were also identified, such as LTR, MBS, and TC-rich repeats (Fig. 7). These results further suggest the probable function of MsMYB genes in response to abiotic stresses.

In the present study, the predictions of protein–protein and protein-DNA interactions were performed to identify some candidate genes that putatively interact with the MsMYB genes. As a result, 170 predicted protein–protein interactions were obtained (Table S8). Of these predicted protein–protein interactions, one interaction between the MsICE1 (MSAD264134) and MsMYB184 protein was predicted, and the interaction of their homologous genes (AtICE1 and AtMYB15) in A. thaliana has been reported (Agarwal et al., 2006). In A. thaliana, the AtMYB15 protein interacts with AtICE1 and binds to Myb recognition sequences in the promoters of CBF genes. The myb15 mutant plants show increased tolerance to freezing stress, whereas its overexpression reduces freezing tolerance. We found that the expression of MsMYB184 was inhibited during cold treatment in this study (Fig. S7), which indicated that this gene might protect alfalfa from cold stress. Additionally, some protein–protein interactions in A. thaliana have also been reported (Mukhtar et al., 2011; Popescu et al., 2009), and the expression levels of their homologous genes under stress conditions were highly correlated, such as MsMYB253 and MSAD220095, MsMYB195 and MSAD211058, and MsMYB204 and MSAD320162 (Fig. 11). A previous study has shown that C/TNGTTA/G were the MYB-core motifs, which are abundantly present in 1-FEH promoters, and CiMYB5 displayed co-expression with its 1-FEH target genes in response to different abiotic stresses and phytohormone treatments (Wei et al., 2017). Moreover, MdMYB23 was confirmed as being able to directly activates the expression of MdCBF1 and MdCBF2 by binding to their promoters, and the transcription of MdANR was also activated by MdMYB23 to promote the biosynthesis of proanthocyanidin, which facilitates ROS scavenging to further enhance cold tolerance (An et al., 2018). In the promoters of MdCBF1, MdCBF2, and MdANR, the MYB-core motifs (C/TNGTTA/G) were also present. In this study, a total of 914 protein-DNA interactions were predicted, and only one homologous gene promoter of interacting genes in the M. truncatula did not contain any MYB-core motifs (C/TNGTTA/G) (Table S9). In order to verify the reliability of these predicted protein–protein interactions, five predicted protein–protein interactions were selected (Table S2), and the interactions between MsMYB043 and MSAD320162, MsMYB253 and MSAD320162, and MsMYB253 and MSAD308489 were confirmed by a yeast two-hybrid system (Fig. 12). The remaining two predicted protein–protein interactions didn’t show a significant relationship, which may require further verification. These results indicate that the predicted interaction genes may interact with MYB genes in alfalfa and play important roles during abiotic stresses.

Conclusions

In order to comprehensively analyze MYB genes in alfalfa, a BLASTP search was performed using 168 A. thaliana, 430 Glycine max, 185 M. truncatula, and 130 O. sativa MYB protein sequences as queries. As a result, a total of 265 non-redundant MYB sequences were obtained and named MsMYB001 to MsMYB265, and these identified MsMYB genes were classified into 12 different subfamilies. Analysis of the physio-chemical properties, motifs and phylogenetic relationships demonstrated that they were mostly similar within the same groups, but greatly differed among different subfamilies. Moreover, GO analysis indicated that most of the MsMYB genes were involved in various biological processes. The expression profiles indicated that most MsMYB genes might participate in the development of alfalfa and respond to abiotic stresses. In addition, a total of 170 protein–protein and 914 protein-DNA interactions were predicted, and the interactions between MsMYB043 and MSAD320162, MsMYB253 and MSAD320162, and MsMYB253 and MSAD308489 were confirmed by a yeast two-hybrid system. This study provides insight into such proteins in alfalfa and a rich resource for subsequent investigations.

Supplemental Information

Supplemental Information 1 Functional annotation (gene ontology) of MsMYB proteins.

Click here for additional data file.

Supplemental Information 2 Schematic representation of motifs identified among 265 MsMYB proteins using the MEME motif searching tool.

Motifs are indicated by different colors. The order of the motifs corresponds to the position of the motifs in individual protein sequences.

Click here for additional data file.

Supplemental Information 3 Hierarchical clustering of interaction gene expression profiles during cold treatment.

These interaction genes were predicted by protein-DNA interaction, and “CK” indicates 0 h. The expression levels of these interaction genes with the absolute value of fold change ≥ 10 and the correlation coefficient between MsMYB genes and its interaction genes was more than 0.8. Microarray data were obtained from the reported study in alfalfa (Zhou et al., 2018).

Click here for additional data file.

Supplemental Information 4 Hierarchical clustering of interaction gene expression profiles during ABA, drought and salt treatments.

These interaction genes were predicted by protein-DNA interaction, and “CK” indicates 0 h. The expression levels of these interaction genes with the absolute value of fold change ≥ 10 and the correlation coefficient between MsMYB genes and its interaction genes was more than 0.8. Microarray data were obtained from the reported studies in alfalfa (Luo et al., 2019a, 2019b).

Click here for additional data file.

Supplemental Information 5 Correlation analysis between the expression patterns of MsMYB genes and their interaction genes during abiotic stresses.

These interaction genes were predicted by protein-DNA interaction. The expression levels of these interaction genes with the absolute value of fold change ≥ 10 and the correlation coefficient between MsMYB genes and its interaction genes was more than 0.8.

Click here for additional data file.

Supplemental Information 6 Five tested yeast co-transformants and its negative controls grew on SD/-U-H-T medium.

Click here for additional data file.

Supplemental Information 7 Expression analysis of MsMYB184 during cold treatment, according to qRT-PCR and RNA-seq.

White bars represent the relative expression levels determined by qRT-PCR (left y-axis). Black bars indicate the transcript abundance change based on the Fragments Per Kilobase per Million Fragments mapped (FPKM) values of the RNA-Seq analysis (right y-axis). Error bars indicate standard errors of the means (n = 3). “*” indicate significance at the 0.05 level.

Click here for additional data file.

Supplemental Information 8 Primers used for qRT-PCR analysis.

Click here for additional data file.

Supplemental Information 9 Primers used for yeast two-hybrid assays.

Red letters indicate enzyme cleavage sites. “Ms gene 2” are interaction genes of the “Ms gene 1.”

Click here for additional data file.

Supplemental Information 10 Summary of the MYB transcription factor genes in alfalfa.

Click here for additional data file.

Supplemental Information 11 Summary information for alignment of the predicted MsMYB genes between a previous study and our research.

Click here for additional data file.

Supplemental Information 12 Functional annotation (gene ontology) of MsMYB proteins.

Click here for additional data file.

Supplemental Information 13 Gene ontology annotations for 45 MYB genes from Arabidopsis thaliana, Lotus japonicas, and Medicago truncatula.

Click here for additional data file.

Supplemental Information 14 MEME motif sequences in predicted MsMYB proteins.

Click here for additional data file.

Supplemental Information 15 Summary information for the 170 protein–protein interactions.

The expression levels of these interaction genes with the absolute value of fold change ≥ 2. The “At gene 2” and “Ms gene 2” are interaction genes of the “At gene 1” and “Ms gene 1,” respectively. The “At gene 1,” “Ms gene 2,” “Mt gene1,” and “Mt gene2” are homologous genes of the “Ms gene 1,” “At gene 2,” “Ms gene 1,” and “Ms gene 2,” respectively.

Click here for additional data file.

Supplemental Information 16 Summary information for the 914 protein-DNA interactions.

The expression levels of these interaction genes with the absolute value of fold change ≥ 2. The “At gene 2” and “Ms gene 2” are interaction genes of the “At gene 1” and “Ms gene 1,” respectively. The “At gene 1,” “Ms gene 2,” “Mt gene1,” and “Mt gene2” are homologous genes of the “Ms gene 1,” “At gene 2,” “Ms gene 1,” and “Ms gene 2,” respectively. The MYB-core motifs (C/TNGTTA/G) were searched in the upstream sequence (2,000 bp) of the homologous genes of interacting genes in M. truncatula.

Click here for additional data file.

Supplemental Information 17 Correlation analysis between the expression patterns of MsMYB genes and their interaction genes during abiotic stresses, which were obtained by the prediction of protein–protein interactions.

The expression levels of these interaction genes with the absolute value of fold change ≥ 2.

Click here for additional data file.

Supplemental Information 18 Correlation analysis between the expression patterns of MsMYB genes and their interaction genes during abiotic stresses, which were obtained by the prediction of protein-DNA interactions.

The expression levels of these interaction genes with the absolute value of fold change ≥ 2.

Click here for additional data file.

Supplemental Information 19 Comparative analysis between the MsMYB and MtMYB genes.

The MtMYB genes are homologous genes of the MsMYB genes.

Click here for additional data file.

Supplemental Information 20 Raw data for Fig. 8.

Click here for additional data file.

We are grateful to Yuguo Wu and Jie Liu for help with the sample collection.

Additional Information and Declarations

Competing Interests

Author Contributions

Data Availability

The authors declare that they have no competing interests.

Qiang Zhou conceived and designed the experiments, performed the experiments, analyzed the data, contributed reagents/materials/analysis tools, prepared figures and/or tables, authored or reviewed drafts of the paper, approved the final draft.

Chenglin Jia performed the experiments.

Wenxue Ma performed the experiments, analyzed the data, prepared figures and/or tables.

Yue Cui performed the experiments, analyzed the data, prepared figures and/or tables.

Xiaoyu Jin contributed reagents/materials/analysis tools.

Dong Luo contributed reagents/materials/analysis tools.

Xueyang Min contributed reagents/materials/analysis tools.

Zhipeng Liu conceived and designed the experiments, authored or reviewed drafts of the paper, approved the final draft.

The following information was supplied regarding data availability:

Data is available at NCBI SRA: SRR7091780–SRR7091794 (cold treatment) and SRR7160313–SRR7160357 (ABA, drought and salt treatments).

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
