# Peer review of "MYB transcription factors in alfalfa (Medicago sativa): genome-wide identification and expression analysis under abiotic stresses"

_PeerJ, doi:10.7717/peerj.7714_

## Round 0.1 · original submission · Major Revisions

As you will see from the reviewer´s comments, your study is very much appreciated but requires a number of clarifications or changes. Please address each point step by step in the revised version. Please pay particular attention to address reviewer 1´s comments on inconsistencies between RNAseq and RT-qPCR data and to clearly distinguish between results from YOUR study and those from other studies. Also, reviewer 2´s comment on comparing M. sativa with M. truncatula is of importance.

Reviewer 1 ·

Basic reporting

#1 Abstract:
The qPCR experiment is not mentioned. Please include it. It is the only experimental result in this manuscript. It comprises conclusive and inconclusive results as compared to a previous RNAseq project and should be discussed as well in more depth (see additional comments at “validity of findings”).
#2
Line 103-105: This goes beyond the depth of this study. The authors should tune down this sentence e.g. to “Results from this study will provide a resource for potential applications of these genes for the improvement of alfalfa tolerance against abiotic stresses.”

Words / nomenclature and sentence structure / readability
#3
Please write out the abbreviation MYB once.
#4
Line 475-482: Please consider using Arabidopsis thaliana (or A. thaliana) instead of Arabidopsis.
#5
Line 392-395. Is renaming required? Apart of this, these lines should appear earlier in the manuscript.
#6
Line 20 and 46: Please consider to split the sentence in two, e.g.: ”… in the world. Its survival…” and e.g.: ”… of the country. Freezing …”
#7
Line 64: The sentence is incomplete. Maybe add “determination” before the bracket.
#8
Line 252ff: I do not find how the presented data contributed to “further understand” the nature of the MsMYB proteins as it is not discussed further.
#9
Line 288 and 482: Please avoid using meanwhile. Do you want to point to another publication dividing the 37 MtMYB proteins in 12 subgroups while their study was ongoing? If so, the citation is missing. Please also delete it in line 482.
#10
Line 374: replace “genes that interact” with “candidate genes that putatively interact”
#11
Line 378ff: Without experimental evidence, please avoid to write about protein-protein and protein-DNA or genetic interactions without unambiguously naming them predicted. Some suggestions: “predicted” instead of “obtained”, “predicted protein-protein interactions”, “predicted protein-DNA interactions”, “its putative interactor’s genes”
#12
Line 476: Adjust “genes that interact“ in a way as suggested before (see related results part).
#13
Line 409: Delete “All”.
#14
Line 426: Insert “conserved” between “six” and “Trp”.
#15
Line 479: MYB15 is the homologous gene for MsMYB184? Please support the reader and unambiguously link the homologous genes/proteins.
#16
Line 487-490: These lines read as if the direct activation of a promoter by a MYB factor can be concluded from the mere presence of a MYB-core motif in this promoter. Please rephrase.

References, Background, introduction
#17
E.g. in line 112: Please cite for the used BLASTs in this manuscript. Here, for example, the named website www. alfalfatoolbox.org refers to Altschul et al., 1990. Please also identify the correct citations for other parts of this study using BLAST including local BLAST.
#18
General: Used web-resources and webtools.
It is required that the authors check all used resources again carefully for potentially overseen required citations and correct for this.
To give an example: e.g. line 156-164 and 316-318: Please read the terms of use as provided by a statement preamble prior to access to the respective website of the Nobel Research Institute and ensure that all required measures allowing you to use and publish the respective data were met. The providers of this web resource request “Prior to publication of the analyses, to accurately and completely cite the pre-publication data, including any applicable analysis as “The Alfalfa Breeder’s Toolbox and Gene Expression Atlas, Maria J. Monteros, Patrick Zhao, Xinbin Dai and Chunlin He. Annotation with CADL Genome v1.0. https://www.alfalfatoolbox.org””
Some of the available datasets are unpublished and it is requested to follow specific terms of usage. I assume that you used only the data set on B47 from the CALD-Gene Expression Atlas here which is published, but not cited, in the current version of your manuscript (O’Rourke et al., 2015).
This also holds true for all used (web)tools. Naming the respective website is not sufficient. Please add the required citations.
#19
Line 60: Please chose a more appropriate reference for the general classification of the MYB family which occurred long before the mentioned paper.
#20
Line 61-62: Please add a citation for this comparison.
#21
In line 73, “lowest level” and “most stable” seem to be out of context here. Please improve this sentence.
#22
Line 65: In the following the authors provide examples for processes for which data is analyzed in this manuscript and qPCR experiments are conducted. At this point of the manuscript, the choice appears arbitrary. The authors could bridge here with the following or something similar and continue with the chosen examples adding an examples for ABA: “MYB transcription factors act for examples in response to cold temperatures, drought, salt stress and ABA.”
#23
Line 68-75: Please adjust these lines together with lines 76-81. In these lines, examples for MYB factors and their function in abiotic stress responses are given and interrupted by the sentence “In fact, …” in line 76. Restructuring by conditions and providing some recent examples per condition might improve this part. Moreover, I do not see that the use of “however”, “moreover” and “in addition” is needed in these lines.
#24
Line 432-447: These examples and those in the following are rather introductory or suited for a review and should be selected for their fit for this study. It would be beneficial if the authors could report on homologous genes which were under investigation here.

Figures
#25
Figure 1: Describe A-N in the legend.
#26
Figure 2: Exchange “asterisks” and “triangles” in the legend.
#27
Figure 3: Describe what is represented by each red dot.
#28
Figure 4: Describe A-H here and in the following figures in a way avoiding the reader might think A in this figure groups the same MYBs as in one of the following figures.
#29
Figure 5: Define “CK” here and in following and supplemental figures.
#30
Figure 7: Consider changing “the cis-element number of MsMYBs.” to “the number of cis-elements in the analyzed promoter region of the indicated MsMYBs.”

Data
#31
Availability of used data.
Section in this manuscript: Line 167-172 and declarations:
The mentioned data for SRR7091780~7091794 is not accessible on Mai, 15th, 2019. Moreover, in the materials and methods section, it is described that the results of the related sequencing projects are deposited under the aforementioned SRR IDs and that these are results of previous studies. Therefore, the data should not be linked in the declarations with this article. The reference in the materials and methods section should remain but authors need to ensure that these resources are accessible which they should be as the previous studies were published with a reference to the same repository and SRR ID. Previous research: Luo et al., 2019a; Luo et al., 2019b; Zhou et al., 2018 with SRR7091780~7091794 and SRR7160313~7160357.

Experimental design

#32
Line 126, 166, 177: It is required to unambiguously differentiate between in silico analysis and experimental analysis. Towards this end, please include “in silico” in the respective headers in the Materials & Methods section and also in the other sections.
#33
Line 466. Please change from: “during multiple abiotic stresses” to “determined in previous studies during multiple abiotic stresses was analyzed.” or similar.
#34
Line 482-483: As the authors used various data of previous studies it is important to specify which finding was experimentally achieved in this study and which finding is a result from analyzing data from previous studies. This occurs repeatedly in the manuscript and should be specified. In the indicated lines, for example, the authors did not “find” that MYB184 is “inhibited during cold” as a results of this study but extracted the results from previously conducted studies. MsMYB184 was also not among the MsMYB factors tested in the qPCR experiment on cold treatment conducted in this study.
#35
Figure legends: Please give information about the respective underlying data for each figure wherever not given so far.
qPCR experiment
#36
Line 364: How did you chose the 14 MYBs? Later, in the discussion, you write that they were randomly selected. How was this done? Can you specify in line 363 what “these” means? Are the 14 tested MYBs a subset of “these” and does “these” relate to the MYB factors anlysed for cis-regulatory elements and therefore belonging to the set of stress responsive MsMYB gene promoter?
#37
E.g. line 209 and 203-206: In this study whole seedlings and root tips were used. Can you motivate the use of the root tips beyond the comparability with the previous RNAseq project of the lab? Please also improve the description and specify for which experiment shown in this study which material was used (seedling or root tips).
#38
Line 218-222: For the primer design: Did you use a genome/transcriptome to avoid off-targets? If so, can you please state which one was used (your lab’s data using Zhongmu No. 1 or e.g. B47 data). As different splicing forms might occur under different treatments, it would also be interesting to know if the used data originated from a control experiment or specific treatments. How did you test the primers? Did you control for primer dimers in the actual experiments using melt curves and did this for all treatment conditions results? Were efficiency tests conducted?
#39
Did you test - or please cite if already published – if one reference gene is sufficient and that the chosen reference gene is not changed under the experimental conditions and therefore qualifies as a reference gene for all conditions.
#40
Line 367-368: Is the induction significant for all genes? See also line 469: What does “markedly” mean in terms of statistics. The significant difference of treatment versus controls could be marked in the figure to visualize which statements can be drawn from both studies (RNAseq and qPCR).
#41
Figure 8: Which sample type was used for which experiment (root tips or whole seedling)? Why did the authors not test MsMYB184 for the cold treatment? The MsMYB184 homolog from A. thaliana and the homolog of its predicted interaction partner are the only pair obtained from the predicted interaction approach for which experimental data from previous studies is discussed by the authors in response to cold.

Validity of the findings

#42 Line 475-482: This is really an interesting finding and interesting background from previous studies on homologous genes and proteins in A. thaliana. Is this the only case for similar data from homologous genes or proteins?
#43 Line 401-406: These are results. Which two studies do the authors relate to here? It can not be this study and Dong et al., 2018 as the manuscript under review does not contain a sequencing experiment.
#44
Line 447-449: Please rephrase. This is not a result of the manuscript under review.

#45
Inconclusive results.

The qPCR results highlight a problem and points to one of the major points of this manuscript which lacks significant discussion: the inconsistency of some RNAseq results and the qPCR results both in terms of trend and more often in terms of differing fold changes as compared to the control.

Here are three possible scenarios:
In scenario one, the RNAseq data is limited and incorrect for some fold changes maybe due to current limitations in the alignment. If this is true, analysis using this type of data and generating heat maps and clustering based on the quantitative data is little informative and might even be misleading.

In scenario two, correctly conducting a qPCR experiment is limited e.g. due to missing information preventing an accurate primer design avoiding off-targets of the same or of different size. Another option might be that the reference gene is not suited for the conditions and tissue under investigation. qPCR experiments measure the steady state levels of a specific transcript (therefore “transcript level” is correct in contrast to “expression”, which is widely used). Depending on the choice of primers, splicing might also have an effect on the detected transcript levels. More reasons for the deviations might be found and should be discussed.

In a third scenario, the experimental conditions are not exactly the same and a factor not considered so far differed between the previous RNAseq experiment and the qPCR experiment like the timepoint within a day when the samples were taken, differences in RNA extraction or cDNA synthesis or differences in the growth conditions of the parental plants as this might affect seedling traits.

However, I would like to point out that the error bars indicate that with the respective available material the experiments were convincingly executed and there is no doubt that these results were obtained.

As this journal supports to report negative and inconclusive results, I suggest to extend the discussion on this aspect. The deviations underline that experiments like RNAseq - and meta-analysis based thereon - require careful verification ideally by different types of experiments but provide a rich and relevant resource for screening for interesting candidates and targets of breeding approaches as done by the authors in this study.

Additional comments

With their manuscript “MYB transcription factors in alfalfa (Medicago sativa): genome-wide identification and expression analysis under abiotic stresses” the authors Qiang Zhou, Wenxue Ma, Yue Cui, Chenglin Jia, Xiaoyu Jin, Dong Luo, Xueyang Min, Yanrong Wang and Zhipeng Liu will provide a resource for potential applications of these genes for the improvement of alfalfa tolerance against abiotic stresses. Moreover the discussion of conclusive but also inconclusive results from the intended verification qPCR experiment of a previous RNAseq experiment will be beneficial for other researchers.

I very much appreciated the efforts of the authors to provide such an extensive compilation of data analysis using previous RNAseq results on alfalfa and data from various other resources. I would like to highlight five points for improving the manuscript from various other points I recorded while reviewing the above mentioned manuscript.

All points are provided in the suggested scheme of the journal divided by three topics above.
1) The improvement of the citation of web resources and web tools.
2) Ensuring the availability of the used data.
3) Improve the unambiguously differentiation of in silico analysis and experimental data as well as results from previous studies and those obtained within the manuscript under review.
4) Improve the presentation and discussion of the qPCR experiment.
5) Extend the discussion and emphasize on the inconclusive results obtained in the qPCR experiments as compared to previous RNAseq results.

·

Basic reporting

I would suggest that a further edit of the manuscript is made to improve the English standard. The language edits would greatly improve the flow of the manuscript.

Experimental design

no comment

Validity of the findings

no comment

Additional comments

The manuscript entitled "MYB transcription factors in alfalfa (Medicago sativa): genome-wide identification and expression analysis under abiotic stresses" by Zhou et al. presents a comprehensive analysis on the MsMYB family genes. The in-silico studies have been carried out systematically The expression profiles indicated that several MsMYB genes might play a crucial role in 
development and abiotic stresses.

However, I would suggest some changes in the current structure of the manuscript. Below please find my specific comments:
In the third para of introduction section, authors mentioned “So far, studies on the MYB gene family have been reported in many plants. A total of 104, 166, 155, 197, 524, and 475 MYB genes were identified and analyzed in Lotus japonicas, M.truncatula, Oryza sativa, Arabidopsis thaliana, Gossypium hirsutum, and Brassica rapa ssp. respectively” From the text it is not clear to which category these MYBs belong. Are these numbers represent all the types of MYB or only R2R3 MYB? Please mention in the revised text clearly.
As Medicago sativa is a close relative of Medicago truncatula. In Medicago truncatula, the genome-wide analysis of R2R3MYB gene family has already been carried out (Zheng et al 2017, J. Int. Agri, 16: 1576-91). Therefore, it would be better to do the comparative analysis of the MsMYB genes with MtMYB genes.
Add one figure showing distribution of MsMYB genes over the chromosome with graphical arrangement of segmental and WGD of these MsMYB.
For phylogenetic analysis why only 37 MtMYB have been selected? For better classification of all the sub groups it would be much worth to add few functionally characterized MYB genes from Medicago, Lotus and Arabidopsis to develop phylogenetic tree. The description of legends of figure 1 is not clear. Please elaborate it and clearly mention according to the biological process of each sub groups from A to N.
Figure quality of S2 and S3 is not clear so improve it in the revised manuscript.
In the supplementary Table S4 and S5, also add the orthologous MtMYB gene along with locus ID. After addition of this authors move it into main text. However authors have flexibility to decide it. Additionally, it would be much worth if the authors can validate transiently Protein-DNA/ Protein-Protein interaction of the few MsMYB. This will further strengthen the quality of the manuscript.

I would suggest that a further edit of the manuscript is made to improve the English standard. The language edits would greatly improve the flow of the manuscript.

---

## Round 0.2 · Major Revisions

Your manuscript has much improved and addressed most of the reviewers´ comments. However, a few issues remain:

1. Regarding the yeast two-hybrid data shown in Figs. 12 and S6: important negative controls are missing that assess autoactivation of the reporter gene by the bait or prey protein, i.e. the combinations bait + empty control of prey and empty control of bait + prey. These negative controls are missing for all interactions tested. Please include the necessary negative controls in all experiments.

2. Please clarify the sentence in the figure legends of several figures: "Microarray data were obtained from the NCBI SRA database of RNA-seq in alfalfa". You did not refer to this database in the Materials section. Please clarify how microarray data are found in an NGS database?

---

## Round 0.3 · accepted · Accept

You addressed my comments in the revised version. Thank you.